# Culture Shock: An Investigation into the Tolerance of Pathogenic Biofilms to Antiseptics in Environments Resembling the Chronic Wound Milieu

**DOI:** 10.3390/ijms242417242

**Published:** 2023-12-08

**Authors:** Justyna Paleczny, Malwina Brożyna, Bartłomiej Dudek, Aleksandra Woytoń, Grzegorz Chodaczek, Marta Szajnik, Adam Junka

**Affiliations:** 1Platform for Unique Models Application, Department of Pharmaceutical Microbiology and Parasitology, Wroclaw Medical University, 50-556 Wroclaw, Poland; justyna.paleczny@student.umw.edu.pl (J.P.); malwina.brozyna@umw.edu.pl (M.B.); bartlomiej.dudek@umw.edu.pl (B.D.); aleks.woyton@gmail.com (A.W.); 2Bioimaging Laboratory, Lukasiewicz Research Network—PORT Polish Center for Technology Development, 54-066 Wroclaw, Poland; grzegorz.chodaczek@port.lukasiewicz.gov.pl; 3Faculty of Medicine, Lazarski University, 02-662 Warsaw, Poland; marta.szajnik@lazarski.edu.pl

**Keywords:** biofilm, wound milieu, in vitro biofilm structure, antiseptics, culture media

## Abstract

Credible assessment methods must be applied to evaluate antiseptics’ in vitro activity reliably. Studies indicate that the medium for biofilm culturing should resemble the conditions present at the site of infection. We cultured *S. aureus*, *S. epidermidis*, *P. aeruginosa*, *C. albicans*, and *E. coli* biofilms in IVWM (In Vitro Wound Milieu)—the medium reflecting wound milieu—and were compared to the ones cultured in the laboratory microbiological Mueller–Hinton (MH) medium. We analyzed and compared crucial biofilm characteristics and treated microbes with polyhexamethylene biguanide hydrochloride (PHMB), povidone-iodine (PVP-I), and super-oxidized solution with hypochlorites (SOHs). Biofilm biomass of *S. aureus* and *S. epidermidis* was higher in IVWM than in MH medium. Microbes cultured in IVWM exhibited greater metabolic activity and thickness than in MH medium. Biofilm of the majority of microbial species was more resistant to PHMB and PVP-I in the IVWM than in the MH medium. *P. aeruginosa* displayed a two-fold lower MBEC value of PHMB in the IVWM than in the MH medium. PHMB was more effective in the IVWM than in the MH medium against *S. aureus* biofilm cultured on a biocellulose carrier (instead of polystyrene). The applied improvement of the standard in vitro methodology allows us to predict the effects of treatment of non-healing wounds with specific antiseptics.

## 1. Introduction

Biofilm is a complex mono- or polymicrobial structure, in which bacterial/fungal cells are embedded within an extracellular matrix (ECM). The ECM consists mainly of proteins, glycoproteins, polysaccharides, and extracellular DNA and serves as a barrier shielding biofilm from unfavorable environmental conditions. It also hinders the action of host immune cells and the penetration of antimicrobials [1,2]. Microbial existence within biofilm fosters the transferring of genes encoding resistance to antibiotics and virulence factors. Due to the mentioned features, biofilm-forming microorganisms demonstrate a 1000 times higher tolerance to antimicrobial agents than their planktonic (free-floating) counterparts [2]. Biofilms are the major factors for persistent hospital infections, such as urinary and biliary tract infections, skin and soft tissue infections, infective endocarditis, and non-healing wounds [3]. Non-healing wounds are diagnosed in 20 million patients each year. Their inefficient treatment may result in limb amputation or severe systemic infection [4]. It was reported that even 78% of non-healing wounds are related to biofilm presence; therefore, only therapies effective against biofilms should be applied in managing non-healing wounds [5]. Locally administered antibiotics are not only ineffective against biofilm structure (because of their poor penetration through the matrix) but also may induce microbial resistance [6]. Therefore, the application of modern antiseptics (frequently in combination with surgical debridement, maggot therapy, and antimicrobial dressings) is highly recommended in non-healing wound treatment. Antiseptic agents display a broad spectrum of antimicrobial and antibiofilm action, being simultaneously non-toxic to skin tissue [7]. They do not target specific cell sites; therefore, the risk of the development of resistance to them is significantly lower than in the case of antibiotics. Antiseptics used the most frequently in the therapy of non-healing wounds are polyhexamethylene biguanide hydrochloride (PHMB), povidone-iodine (PVP-I), and super-oxidized solution with hypochlorites (SOHs) [8].

PHMB is a polymer composed of varying lengths of PHMB chloride chains. It is noteworthy because polyamines display high dependance of activity in function of molar mass [9]. PHMB acts through the interaction with the microbial membrane, damaging it and eventually leading to cell death. The agent may also impair the DNA structure. PHMB exhibits a broad spectrum of antibacterial activity, which includes Gram-positive, Gram-negative, and intracellular ones. Numerous reports demonstrate its high activity against antibiotic-resistant strains such as methicillin-resistant *Staphylococcus aureus* (MRSA) and vancomycin-resistant *Enterococcus* (VRE), fungi, viruses (*Human Immunodeficiency Virus* (HIV-1), *Herpes Simplex Virus* (HSV)), and protozoa. PHMB do not possess sporicidal activity. The antiseptic effectively eradicates biofilm and prevents its formation.

The antiseptic’s antimicrobial activity is not reduced in human fluids (blood, albumin) and tissues. Moreover, PHMB may act synergistically with particular antibiotics. Its other important features are good local tolerability, negligible absorption from skin and wound, low risk of allergy, and no genotoxic effect [8,10].

SOHs is a mixture of hypochlorites such as sodium hypochlorite, hypochlorous acid, and hypochlorite. The hypochlorite mode of antimicrobial action is based on the oxidation of cell structures, which results in their degradation and cell death. Although the antiseptic is non-toxic for humans, its antimicrobial effectiveness in clinical use is disputable. Reports indicate that SOHs possesses antibacterial, antifungal, and antiviral activity. However, the antimicrobial effect is achieved in highly concentrated solutions, while numerous available products are significantly less concentrated [11,12,13,14].

PVP-I is not only effective against bacteria and fungi but also against their vegetative and spore forms and viruses. The antiseptic releases free iodine molecules, which easily penetrate the microbial cell wall. Iodine forms irreversible binding with proteins, fatty acids, and pyrimidine bases. It results in enzymatic inactivation, disruption of DNA structure, and cell membrane damage, which leads to cell death. PVP-I is recommended for skin, mucosa, oral cavity, and wound antisepsis. Iodine is highly absorbed from the skin and influences the thyroid. Therefore, the antiseptic is contraindicated for premature infants, elderly people, and pregnant women. There is no evidence for clinically relevant microbial resistance to PVP-I [15].

Credible assessment methods need to be applied to reliably evaluate antiseptics’ in vitro activity reliably. Numerous research studies indicate that the medium for biofilm culturing should reflect the conditions present at the site of infection. Crucial biofilm features (e.g., spatial structure, metabolic activity) may differ depending on the used medium, which translates to biofilms distinct tolerance to antimicrobial agents [16,17]. Standard microbiological media, such as Mueller–Hinton broth (MH) and Tryptic Soy Broth (TSB), contain high concentrations of proteins, polysaccharides, and amino acids—components not present so abundantly in wound fluid. In turn, Kadam et al. proposed an IVWM (In Vitro Wound Milieu) medium to be applied for the tests evaluating the efficacy of antimicrobials used for the therapy of wound infections. IVWM is composed of serum, elements of cell matrix, and host biochemical factors released in response to tissue damage or microbial invasion. The high resemblance in serum and wound fluid’s biochemical and nutrient profiles was previously demonstrated. Moreover, in the bed of an infected wound, aggregated biofilm cells bind to elements of the host matrix. Therefore, using IVWM in an experimental setting allows one to mimic the actual state of an infected, non-healed wound. Based on in vitro results of antiseptics’ antimicrobial efficacy, recommendations for their in vivo applicability are being developed. Thus, thanks to providing the conditions reflecting the wound milieu in in vitro antiseptic studies, the results obtained are more reliable and allow the choice of appropriate agents, which translates to higher chances of clinical success [18].

In the present study, we cultured *S. aureus*, *S. epidermidis*, *P. aeruginosa*, *C. albicans*, and *E. coli* biofilms (which are major causative factors of wound infections) in the MH or IVWM medium. We characterized and compared biomass level, metabolic activity, thickness, and live/dead cell distribution of biofilms cultivated in an MH medium to the ones grown in an IVWM medium. Subsequently, we evaluated the biofilms’ tolerance to such antiseptics as PHMB, SOH, and PVP-I in both media. We also performed control analyses on the planktonic cells of the tested microbes. Additionally, the antiseptics’ antibacterial effectiveness was assessed against biofilm formed on three-dimensional bacterial cellulose carriers soaked with the MH or IVWM medium.

## 2. Results

During the initial stage of the investigation, the ability of bacterial and fungal strains to produce biofilms was scrutinized (Figure 1). The crystal violet staining technique was employed to quantify the biomass of the tested strains. The findings indicated that both *S. aureus* and *S. epidermidis* exhibited a significantly greater amount of biomass in the IVWM (In Vitro Wound Milieu) as compared to MH (Mueller–Hinton) medium (Figure 1A). However, it is noteworthy that no statistically significant differences were observed for other types of microbial species. In the conducted study, the metabolic activity of microbial species was assessed through the utilization of tetrazolium reduction method for bacteria (Figure 1B) and the resazurin method for fungi (Figure 1C). The physical appearance of biofilms was found to be highly dependent on the type of medium utilized with noticeable variations observed (Figure 1D). Upon analyzing the obtained data, it was observed that microbes cultured in IVWM exhibited a notably higher level of metabolic activity as compared to MH. This suggests that IVWM may provide a more favorable environment for microbial growth and metabolic processes. Further research may be necessary to fully understand the intricacies of biofilm formation under various conditions.

In the next stage of the experiment, the biofilm was meticulously observed via fluorescence microscopy after being dyed with fluorescence staining (Figure 2 and Figure 3). This technique provided an opportunity to ascertain the thickness of the biofilm and the proportion of living/non-damaged to damaged cells in each layer of the biofilm structure. In comparison to MH cultures, the biofilm thickness was generally greater in IVWM for most tested strains. However, there was only a statistically significant difference observed in the case of *C. albicans* (Figure 4). The only exception to this trend was the *P. aeruginosa* biofilm, which appeared slightly thinner in MH. It is worth noting that the IVWM outcomes had a higher standard deviation, suggesting a lower level of replicability. Confocal microscopy with live/dead staining results revealed more non-damaged cells in the biofilm formed in IVWM. This outcome aligns with the results obtained from tetrazolium or resazurin methods, which showed that the cells within the biofilm in IVWM had a higher metabolic activity. It was observed that the number of non-damaged and damaged cells in a biofilm layer varied depending on the culture medium used. In the case of IVWM, the greatest concentration of viable cells was found in the initial 20–30% of the thickness, whereas with MH, they were mostly located in the middle of the biofilm layer.

The next step involved determining the minimum inhibitory concentration (MIC) and the Minimal Biofilm Eradication Concentration (MBEC) of the three antimicrobials that have been tested. The study found that higher minimum inhibitory concentrations of PHMB were observed in IVWM for Gram-negative bacteria *E. coli* and *P. aeruginosa* as compared to MH (Table 1). Conversely, the MICs of PVP-I were found to be higher in IVWM by up to 4-fold in the case of *E. coli* and *C. albicans* and 8-fold in *S. aureus* and *S. epidermidis*. However, the SOHs compound exhibited no growth inhibitory effect on most of the microorganisms tested, irrespective of the culture medium. These findings highlight the variations in the effectiveness of different compounds against various microorganisms and the impact of culture media on their efficacy. It has also been observed that the biofilm of the majority of microbial species exhibit an elevated resistance to PHMB and PVP-I when cultured in the IVWM medium as compared to the MH medium (Table 2). However, there is an exception in the case of *P. aeruginosa*, which displays a two-fold lower MBEC of PHMB in the IVWM medium. The use of SOH-containing agent was found to be less effective in both types of media than PHMB and PVP-I. The difference between MBEC values in both media was observed only for *C. albicans*. Furthermore, it was found that the *S. epidermidis* growth was insufficient to determine the minimum concentration of biofilm eradication by any antimicrobial agents tested.

Finally, the antimicrobial efficacy was tested against biofilm formed on three-dimensional bacterial cellulose carriers previously soaked in both tested media. In this approach, PHMB and SOHs were chosen due to their respective high and low efficacies in the abovementioned methods. After the treatment with PHMB, the viability of cells in biofilm formed in MH was significantly higher than in IVWM, which shows that the eradication was more effective against biofilms formed in IVWM than MH (Figure 5A). The results obtained for SOHs were comparable for biofilm formed in both media. The staphylococcal biofilm was visualized by employing the Richards method. Figure 5B shows the discrepancies in the biofilm morphology depending on the medium in which the bacteria were cultured.

## 3. Discussion

The detrimental role of biofilms in the pathogenesis of infections has been well-documented across a wide range of major human body sites, including the oral cavity, respiratory and alimentary systems, soft tissues, and bones [19]. The development of biofilms in chronic wounds also provides a prime example of the features of this microbial community, making their removal challenging due to altered metabolism, shielding within the extracellular matrix, and the interchange of information via gene transfer and the quorum sensing system [20,21,22].

Therefore, locally acting antimicrobial agents, namely antiseptics intended for use on biofilm-infected chronic wounds, should exhibit extremely high killing activity and a broad spectrum of activity, while remaining non-toxic to eukaryotic cells [7]. The evaluation of antiseptic efficacy is based on in vitro studies using normative methods, which do not fully replicate the wound and biofilm environment, and/or on in vitro models that attempt to encompass both the unique aspects of the chronic wound milieu and biofilm characteristics [23]. One of the crucial components of in vitro setting for testing biofilms’ tolerance to antiseptics is the medium the biofilm is immersed in. It should reflect the wound exudate, the fluid emitted from a chronic wound. It is a mixture of water, proteins, nutrients, immune cells, and waste products (sometimes it can contain blood and then it is referred to as the bloody exudate) [24]. Exudate plays a crucial role in the healing process by keeping the wound moist. In chronic wounds, which fail to heal in a timely manner through the normal stages of healing, the nature and quantity of exudate can change, often resulting in excessive production, altered composition, and impedance of healing [25].

The exudate from chronic wounds contains a variety of components with antimicrobial properties, integral to the body’s natural defense mechanisms against infection and to promote healing, including white blood cells [26], antimicrobial peptides (AMPs) [27], enzymes (lysozyme, lactoferrin, and others) [28], immunoglobulins, or fibronectin [29].

Incorporating all antimicrobial components into the artificial exudate (i.e., medium for in vitro biofilm testing) would be extremely challenging from practical, technical, and economical perspectives, particularly concerning its cellular aspects. It would limit its practicality for routine, standard applications in microbiological laboratories, where large volumes of media are required. The IVWM (In Vitro Wound Milieu) medium, formulated by Kadam et al., represents an effort to include as many crucial components of exudate and the wound bed environment as possible while maintaining affordability [18]. This is achieved by reducing cellular components (such as macrophages and leukocytes) and including more accessible and less resource-intensive elements like Fetal Bovine Serum, lactic acid, lactoferrin, fibrinogen, fibronectin, and collagen.

Therefore, the main question addressed in this research is as follows: Do biofilms cultured in the IVWM medium display different basic features compared to biofilms cultured in Mueller–Hinton medium (commonly used for many diagnostic tests), and as a result, do IVWM-based biofilms exhibit differing tolerance to antiseptics compared to those grown in MH medium? The additional question is whether the same trends concerning level of biofilm biomass, biofilm thickness, and distribution of live/dead cells within its structure occur in different species of wound pathogens tested?

Data presented in Figure 1 indicate that the total level of biofilm biomass, comprising both biofilm cells and the biofilm matrix, was significantly higher in biofilms formed by Gram-positive *S. aureus* and *S. epidermidis* species in the IVWM medium compared to Mueller–Hinton (MH); however, it was comparable for Gram-negative *E. coli*, *P. aeruginosa*, and *C. albicans*. This difference observed in the two staphylococcal species is intriguing, especially considering that the MH medium offers a higher nutrient content and lacks potentially antimicrobial components, unlike IVWM. In contrast, the level of metabolic activity in all biofilms was higher in IVWM than in MH. This could be attributed to at least two interconnected factors. Firstly, cells in the IVWM medium, sensing stress factors such as lactoferrin, might enhance their metabolic activity to protect themselves and rapidly and efficiently acquire iron—a necessity does not present in the nutrient-rich, safer MH medium [30]. Secondly, cells in IVWM under stress might invest more energy in producing a protective matrix, a process that would result in higher energy consumption, as measured and detected by Richards method [31].

The Gram-negative *P. aeruginosa* and *E. coli* are renowned for their excessive production of slimy biofilm matrix in virtually all types of in vitro settings, which could account for their biofilm biomass being at the same level in both IVWM and MH media due to the high amount of slime. However, the level of metabolic activity was higher in the IVWM medium. Conversely, staphylococci typically form multicellular adhered aggregates in standard in vitro media (such as MH, TSB (Tryptic Soy Broth), or BHI (Brain Heart Infusion)), which are not necessarily embedded within a strong biofilm matrix [32]. Thus, it could be hypothesized that the inclusion of Fetal Bovine Serum supplemented with lactoferrin and lactic acid in IVWM prompts staphylococci to allocate their energy not only to inner cellular processes but also to the production of an extracellular matrix. This hypothesis provides a plausible explanation for the observed significant increase in staphylococcal biofilm biomass in IVWM compared to the biomass level measured in the MH setting, as illustrated in Figure 1A. The differences highlighted in Figure 1A,B are expected to manifest in the shape and spatial distribution of biofilm-forming cells. To assess the impact of the two different media on these features, fluorescence microscopy was employed. Each biofilm, formed by every pathogen tested, exhibited distinct characteristics in the IVWM versus the MH setting, as demonstrated in Figure 3. A significant observation was that biofilms cultured in IVWM were more aggregated and did not homogeneously cover the growth surface, unlike those cultured in MH. Instead, they formed bubble-like, dense structures, reminiscent of cellular aggregates typically seen in clinical specimens, such as tissues [33]. This suggests that biofilms grown in IVWM more closely resemble the nature of in vivo wound biofilms. This similarity is likely due to the microbial growth-limiting factors present in IVWM, such as serum compounds, lactoferrin, and lactic acid. These components may induce a sense in the biofilm-forming cells that they are in harsher conditions, leading to aggregation as a defensive mechanism.

The biofilms formed in the IVWM medium were observed to be thicker than those formed in the MH medium (Figure 2 and Figure 4). Importantly, an increasing number of researchers are drawing attention to the fact that the conditions provided in vitro by standard microbiological media are overly favorable [34,35,36,37]. These media were originally designed to promote the growth of microbial cells, as a higher cell count aids the diagnostic process, but they do not replicate the often harsh environments found in human body niches, such as chronic wounds. The hypothesis is that microorganisms cultured under such conditions may not activate, or at least not fully activate, the expression of genes responsible for differentiation in biofilm structure or for defense mechanisms. Consequently, the biofilm derived from cultures in standard media may more closely resemble a bulk or pile of cells adhered to an abiotic surface (like polystyrene) rather than a protective structure capable of withstanding challenges from the immune system and the activity of antimicrobial agents. Notably, the ratio of live to dead cells in biofilms cultured in IVWM did not differ significantly from those formed in MH (Figure 2). However, a certain representation of the latter mentioned cells was present throughout the entire biofilm structure. This observation is consistent with data previously presented by our team and other researchers.

Recognizing that biofilms cultured in IVWM were generally more metabolically active, thicker, and more aggregated compared to those cultured in MH, the next phase of our investigation involved comparing the Minimal Inhibitory Concentrations (MICs) of three clinically applied antiseptic agents—PHMB, SOH, and PVP-I—against these differently cultured biofilms. This comparison was conducted using the standard 96-well plate test. The introduction of antiseptics was a crucial step in this research, notably adding another variable, as each antiseptic employed has a different mechanism of action, which could potentially influence the observed phenomena in various ways [7,12,38].

As indicated in Table 1, planktonic pathogens cultured in IVWM and exposed to PHMB demonstrated increased tolerance for *E. coli* and *P. aeruginosa*, while tolerance levels for *Staphylococci* and *Candida albicans* remained the same compared to the aforementioned antiseptic. This suggests PHMB’s effective killing ability towards cells cultured in a medium containing antimicrobial components. Cells in such an environment may develop altered cell walls/membranes or induce proton pumps to efflux antimicrobials, thereby preventing cellular destruction [39,40]. The precise comparison of SOH’s MIC in two media was not determinable, as all pathogens cultured in IVWM were tolerant to this antiseptic within tested ranges of concentrations. However, in the sole case where a difference was measurable, the MIC of SOHs in MH for *S. epidermidis* was lower (more favorable) than the MIC for *S. epidermidis* cultured in IVWM. When exposed to the denaturing PVP-I antiseptic agent, cells cultured in IVWM exhibited significantly increased tolerance, with four out of the five tested species showing enhanced resistance to this antiseptic. Overall, the results demonstrate that cultivation in a medium resembling the fluid of chronic wounds leads to an adaptive response in microbial cells, as components of biofilm, resulting in elevated tolerance to antiseptics. This increased tolerance was generally observed regardless of the type of antiseptic applied and its specific mechanism of action.

In the primary focus of our investigation, we assessed the Minimal Biofilm Eradication Concentrations (MBECs) of the three previously mentioned antiseptic agents (PHMB, SOHs, and PVP-I) against biofilms cultivated in either the IVWM or MH medium, as detailed in Table 2. This step introduced an additional variable into the experimental framework, related to the spatial distribution of cells within the biofilm, the presence of the matrix, and altered metabolic levels. The results obtained indicate that biofilms formed by various species and cultivated in IVWM generally exhibited higher tolerance to the applied antiseptics. This increased tolerance was particularly notable in the case of *C. albicans* biofilms exposed to PHMB, where tolerance levels were a few dozen times higher and typically about twice as high for other antiseptics and pathogens. The only significant exception to this trend was observed in the biofilms of *P. aeruginosa*. However, we are convinced that this deviation is more likely attributable to the well-known challenges associated with processing biofilms formed by this pathogen in 96-well plate models, rather than being indicative of any phenomena related to the activity of the antiseptics [41,42,43].

In a final demonstration of our concept, we conducted an analysis with the prevalent pathogen *S. aureus* and two antiseptics, PHMB and SOHs, which have different modes of action [12]. This analysis was distinct in that not only did the medium resemble chronic wound exudate, but also the standard polystyrene abiotic surface was replaced with a soft, porous, 3D bio-nanocellulose structure mimicking wound tissue [44], as shown in Figure 5. The results indicated that PHMB was able to reduce approximately half of the staphylococcal biofilm formed under these conditions, whereas SOHs displayed negligible to no antibiofilm activity. This finding is consistent with data reported by our team and others. Interestingly, when contrasted with the data in Table 2, where SOHs was relatively effective in reducing staphylococcal biofilm, it appears that the introduction of a porous cellulosic mesh, which allows microorganisms to penetrate through (similarly to necrotic layers in chronic wounds [45]), significantly diminishes the efficacy of SOHs in exhibiting killing activity. Above shows the importance of application of different in vitro models to obtain the full insight into possible activities of applied antiseptic agents towards pathogenic biofilms in vitro and to obtain full insight before attempting to translate such data on the clinical conditions [17,46,47]. This comprehensive approach ensures that the nuances and complexities of biofilm behavior and antiseptic efficacy in different environments are adequately considered, leading to more accurate and applicable results in clinical settings.

In our research, we employed a range of in vitro models to compare the characteristics of pathogenic biofilms cultured in conditions that mimic the chronic wound environment with those grown under standard microbiological culture conditions. We found that biofilms developed in the former conditions are thicker, more aggregated, and display greater metabolic activity. These attributes contribute to the observed increased tolerance of IVWM-cultured biofilms to various antiseptics. This demonstration could be an important step in predicting the actual activity of antiseptics in clinical settings. On the other hand, we acknowledge that our research would benefit from the inclusion of not only reference strains but also many clinical strains. This would allow for an assessment of intra-species variability in response to antiseptic agents. Additionally, the application of microscopic techniques, such as transmission electron microscopy, could be instrumental in measuring potential changes in cell wall thickness between IVWM and MH settings. Spectrometric methods could also be employed to evaluate the amount and changes in the extracellular matrix. Research in this field is an ongoing process, where each answer leads to new questions. Nonetheless, by providing significant evidence that microbial biofilms may be more tolerant to antiseptics than previously thought, based on data from cultures performed in standard microbiological media, we are moving closer to understanding not just the behavior of biofilms in chronic wounds but also to developing a set of adequate in vitro models that could potentially predict the effects of treatment with specific antiseptic agents on biofilms formed by particular pathogens in chronic wounds.

## 4. Materials and Methods

### 4.1. Microorganisms and Culture Conditions

The research was performed on four bacterial- and one fungal-type strains from American Type Culture Collection (ATCC, Manassas, VA, USA) or Polish Collection of Microorganisms (PCMs) and were a part of the Strains Collection of the Department of Pharmaceutical Microbiology and Parasitology, Medical University of Wroclaw, Poland. The tested strains were as follows:Staphylococcus aureus ATCC 6538;Staphylococcus epidermidis PCM 2118;Pseudomonas aeruginosa ATCC 27853;Escherichia coli ATCC 25922;Candida albicans ATCC 10231.

All strains were cultured in two compositionally different broths for experimental purposes. The reference medium was Mueller–Hinton broth (MH; Biomaxima, Lublin, Poland). An In Vitro Wound Milieu (IVWM) was applied to mimic wound infection site and was prepared according to Kadam et al. procedures [18]. IVWM composites consisted of 70% of Fetal Bovine Serum (FBS; Biowest, Nuaillé, France, cat. No. S181H), lactic acid 11 mM (from a 11.4 mM stock solution in water) (Sigma-Aldrich, St. Louis, MO, USA, cat. No. W261114), lactoferrin 20 µg/mL (from 2 mg/mL stock solution prepared in PBS) (lactoferrin human, Sigma-Aldrich, St. Louis, MO, USA, cat. No. L4040), fibrinogen 200 mg/mL (from 10 mg/mL stock solution prepared in pre-warmed water) (fibrinogen from human plasma, Sigma-Aldrich, St. Louis, MO, USA, cat. No. F3879), fibronectin 30 µg/mL (from 1 mg/mL stock solution prepared in pre-warmed saline (Stanlab, Lublin, Poland)) (human plasma fibronectin, Sigma-Aldrich, St. Louis, MO, USA, cat. No. FC010), and sterile collagen 10 µg/mL (collagen solution from bovine skin, Sigma-Aldrich, St. Louis, MO, USA, cat. No. C4243). All components, except for sterile FBS and collagen, were filter-sterilized with a 0.22 µm syringe filter (Sungo, Europe, Amsterdam, The Netherlands).

### 4.2. Ability of the Strains to Form Biofilms

#### 4.2.1. Biomass Measurement

Microorganisms were cultured in MH (Mueller–Hinton broth; Biomaxima, Lublin, Poland) or IVWM (In Vitro Wound Milieu). Suspensions of 0.5 McFarland (Densitomat II; BioMerieux, Warszawa, Poland) were obtained from freshly prepared 24 h liquid cultures and geometrically diluted 1000 times in the particular medium. Next, 100 µL of the bacterial/fungal suspensions was added to 6 wells of a 96-well plate (VWR, Radnor, PA, USA) and incubated for 24 h at 37 °C under stationary conditions. After time, the supernatant was gently removed, and the plate was dried for 10 min at 37 °C. Next, 100 µL of 20% (*v*/*v*) aqueous crystal violet solution (Aqua-med, Łódź, Poland) was added for 10 min. All wells were rinsed with 100 µL 0.9% (*w*/*v*) NaCl twice and dried for next 10 min at 37 °C. A total of 100 µL of 30% (*v*/*v*) acetic acid (Chempur, Piekary Śląskie, Poland) was added to each well with stained biofilm, and the plate was shaken for 30 min at 450 rpm (Schuttler MTS-4, IKA, Staufen, Germany). After transferring the color solutions to a new 96-well plate, the absorbance was measured at 550 nm using MultiScan Go Spectrophotometer (Thermo Fischer Scientific, Waltham, MA, USA).

#### 4.2.2. Metabolic Activity Measurement (Richards Method)

The culture under stationary conditions was performed as described above. After the incubation time, the suspension above the biofilm was carefully aspirated. A total of 100 µL of the appropriate dyes, prepared in MH (Mueller–Hinton broth; Biomaxima, Lublin, Poland) or IVWM (In Vitro Wound Milieu), was added to the biofilm wells, and the plate was incubated for 2 h at 37 °C. For bacteria, 0.1% (*w*/*v*) tetrazolium chloride salt (TTC; 2,3,5-triphenyl-2H-tetrazolium chloride, PanReac AppliChem, Darmstadt, Germany) was used, while for *Candida albicans*, 0.001% (*w*/*v*) resazurin sodium salt solution (Acros Organics, Geel, Belgium) was applied. To dissolve the red formazan crystals formed from TTC, the supernatant was discarded, and 100 µL of methanol/acetic acid mixture (9:1, *v*/*v*, POCH, Poland) was added to the wells and shaken for 30 min at 450 rpm (Schuttler MTS-4, IKA, Germany) at room temperature. After transferring the solutions to a new 96-well plate (VWR, Radnor, PA, USA), the absorbance was measured at 490 nm (MultiScan Go Spectrophotometer, Thermo Fischer Scientific, Waltham, MA, USA). In turn, the absorbance of the samples stained with resazurin solution was measured at wavelengths 570 nm and 600 nm.

#### 4.2.3. Confocal Microscopy

A total of 1 mL of bacterial and fungal suspensions at density 0.5 McFarland (Densitomat II; BioMerieux, Poland) diluted 1000 times in MH (Mueller–Hinton broth; Lublin, Biomaxima, Poland) or IVWM (In Vitro Wound Milieu) medium was added to four wells of a 24-well plate (VWR, Radnor, PA, USA) and incubated at 37 °C overnight under stationary conditions. Next, the biofilms were dyed with FilmTracer™ LIVE/DEAD™ Biofilm Viability Kit following the manufacturer’s instruction (Thermo Fischer Scientific, Waltham, MA, USA) and visualized, as described before [12], using a confocal microscope Leica TCS SP8 with a water-dipping objective (Leica Microsystems, Wetzlar, Germany) or using a fluorescence microscope Lumascope 620 (Etaluma, Carlsbad, CA, USA). A laser line (confocal microscopy) at wavelength 488 nm (500–530 nm emission) was applied to visualize SYTO-9, when a wavelength at 552 nm (575–627 nm emission) was applied to visualize propidium iodide (PI) in a sequential mode. Images presented the maximum intensity projections acquired from confocal stacks with 2 µm spacing in the Z dimension. The orange/red color showed damaged cells thanks to the PI staining, and the green color of the SYTO-9 staining represented non-damaged cells. The intensity of each staining was measured with Fiji 2.9.0 (accessed on 1 February 2023) [48].

### 4.3. Antimicrobial and Antibiofilm Activities of Selected Antimicrobials

#### 4.3.1. Antimicrobials

The liquid antiseptics applied in the study were as follows:Polyhexamethylene biguanide hydrochloride (PHMB)—Prontosan^®^ wound irrigation solution (B. Braun, Melsungen, Germany), containing polyhexamethylene biguanide (0.1%), undecylenamidopropyl betaine (0.1%), and purified water.Super-oxidized solution with hypochlorites (SOHs)—Granudacyn^®^ wound irrigation solution (Molnycke, Gothenburg, Sweden) containing water, sodium chloride, hypochlorous acid (0.005%), and sodium hypochlorite (0.005%).Povidone-iodine (PVP-I)—Braunol^®^ skin solution liquid (B. Braun, Melsungen, Germany), containing 7.5% of iodized povidone with 10% available iodine, sodium dihydrogen phosphate dihydrate, sodium iodate, macrogol 9 lauryl ether, sodium hydroxide, and purified water.

All agents were neutralized with a neutralization agent composed of 4% (*v*/*v*) Tween 80 (VWR, Radnor, PA, USA), 3% (*w*/*v*) saponin (VWR, Radnor, PA, USA), 1% sodium dodecyl sulphate (SDS), and 0.4% lecithin in 1 L of distilled water.

#### 4.3.2. Minimal Inhibitory Concentration (MIC)

Geometric dilutions of antimicrobials were prepared in MH (Mueller–Hinton broth; Biomaxima, Poland) or IVWM (In Vitro Wound Milieu) medium in 96-well plates (VWR, Radnor, PA, USA) with a final volume of 50 µL. The highest applied concentrations of compounds were, for PHMB, 0.5 g/L; for SOHs, 0.05 g/L; and for PVP-I, 37.5 g/L. Bacterial and fungal suspensions at a density of 0.5 McFarland (Densitomat II, BioMerieux, Warszawa, Poland), prepared from 24 h cultures in the appropriate medium, were diluted 1000 times. Then, 50 µL of suspension was added to each well in a series of antiseptic dilutions. After measuring absorbance at 580 nm (MultiScan Go Spectrophotometer, Thermo Fischer Scientific, Waltham, MA, USA), the plates were placed on a shaker and incubated for 24 h at 37 °C, 450 rpm (Schuttler MTS-4, IKA, Staufen, Germany). Next, the absorbance was measured again at 580 nm, indicating the well with the lowest substance concentration that inhibited the growth of the pathogens. This concentration was taken as the Minimum Inhibitory Concentration (MIC), which was confirmed using the colorimetric method. To inhibit the antimicrobials activity, 100 µL of neutralizer was added to each well. Dye solutions were freshly prepared in MH or IVWM. A total of 20 µL of 1% (*w*/*v*) tetrazolium chloride salt (TTC; 2,3,5-triphenyl-2H-tetrazolium chloride, PanReac AppliChem, Darmstadt, Germany) was added to the wells with bacterial suspensions, or 20 µL 0.01% (*w*/*v*) resazurin sodium salt (Acros Organics, Geel, Belgium) was added to the wells with fungal suspensions. After the 2 h incubation at 37 °C, the plates were read visually. Thus, the MIC was determined as the concentration of antiseptic in the first non-red-stained well (for bacteria) or the first blue well next to the pink well (for fungi) due to the change in color of the medium, resulting from microbial metabolic activity. Sterility and bacterial growth controls were prepared for each setup. Three replicates were performed for each tested strain.

#### 4.3.3. Minimal Biofilm Eradication Concentration (MBEC)

The biofilm was established from fresh, 24 h cultures in MH (Mueller–Hinton broth; Biomaxima, Lublin, Poland) or IVWM (In Vitro Wound Milieu) medium. The bacterial/fungal suspension at density 0.5 McFarland (Densitomat II, BioMerieux, Warszawa, Poland) was diluted 1000 times in the specific medium. Thus, 50 µL of the culture was added to the wells of 96-well plates (VWR, Radnor, PA, USA) and filled with an extra 50 µL of medium. The plate was incubated under static conditions for 24 h at 37 °C. The following day, the supernatant above the biofilm, formed at the bottom of the wells, was gently removed. Next, 100 µL of geometric dilutions of the antimicrobials, prepared in MH or IVWM medium, was added to each plate. The highest concentrations of antimicrobials were as follows: PHMB—0.1 g/L, SOH—0.01 g/L, and PVP-I—75 g/L. The growth control was provided as a bacterial biofilm with medium, with no antimicrobial added. The sterility control was provided as a culture medium with no bacteria or antiseptics. Next, the plate was incubated for 24 h at 37 °C. After the incubation time, 100 µL of a neutralizer was added to each well for 5 min, and then the supernatant was gently removed. To the wells with bacterial biofilm, 100 µL of 0.1% (*w*/*v*) tetrazolium chloride salt (TTC, 2,3,5-triphenyl-2H-tetrazolium chloride, PanReac AppliChem, Darmstadt, Germany) was added when wells with fungal biofilm were filled with 100 µL of 0.001% (*w*/*v*) resazurin sodium salt (Acros Organics, Geel, Belgium), each prepared in MH or IVWM medium. The plates were incubated for 2 h at 37 °C, and then the MBEC (Minimal Biofilm Eradication Concentration) values were read visually. Due to the color change in the medium resulting from the microbial metabolic activity, the MBEC values for bacteria were determined as the concentration of antiseptic in the first non-red-stained well, while for fungi, the first blue well was next to the pink well.

#### 4.3.4. Cellulose-Based Biofilm (CBB) Model

For the research purpose, bacterial cellulose (BC) carriers were biosynthesized as described in our previous research [49]. Bacterial carriers at diameter 18 mm were soaked in MH (Mueller–Hinton broth; Biomaxima, Lublin, Poland) or IVWM (In Vitro Wound Milieu) medium overnight at 4 °C. Next day, BC carriers were transferred to 24-well plates (VWR, Radnor, PA, USA). The suspension of *Staphylococcus aureus*, at density 0.5 McFarland and diluted 1000 times was prepared in MH or IVWM medium. A total of 1 mL of such suspension was added to each well with BC and incubated for 48 h at 37 °C under stationary condition. After time, the suspension was gently removed, and the BC carriers were transferred to new 24-well plates. A total of 1 mL of undiluted antimicrobial or 0.9% (*w*/*v*) NaCl (for growth control) was added to the BC with bacterial biofilm and left for 15 min at room temperature. Next, 1 mL of neutralizer was added to each well for 5 min. To visualize bacterial biofilm, 1 mL of 0.1% (*w*/*v*) tetrazolium chloride salt (TTC; 2,3,5-triphenyl-2H-tetrazolium chloride, PanReac AppliChem, Darmstadt, Germany) prepared in MH or IVWM medium was added to each well with BC. After 2 h incubation at 37 °C, the solution was gently removed, and 1 mL of methanol/acetic acid mixture (9:1, *v*/*v*, POCH, Poland) was added to the wells and shaken for 30 min at 450 rpm (Schuttler MTS-4, IKA, Staufen, Germany). After transferring the color solutions to a 96-well plate (VWR, Radnor, PA, USA), the absorbance was measured at 490 nm (MultiScan Go Spectrophotometer, Thermo Fischer Scientific, Waltham, MA, USA). The cells’ viability in biofilm (V) was calculated with Equation (1):(1)$$V(\%)=\frac{{Abs}_{T}}{{Abs}_{GC}}\times 100$$
where *Abs_T_* stands for an average of absorbance values obtained for test sample, and *Abs_GC_* stands for an average value of absorbance values obtained for growth control. Two of the three antiseptics, with the highest or lowest biofilm-eradicating activity, PHMB or SOH, respectively, were tested with this method, which was selected based on the results obtained with the previously described techniques. Four replicates were performed for each setup.

### 4.4. Statistical Analysis

The results were analyzed using GraphPad Prism (Version 8.0.1; GraphPad Software Inc., San Diego, CA, USA, www.graphpad.com (accessed on 20 February 2023)). The normality distribution and variance homogeneity were assessed with Shapiro–Wilk and Levene’s tests, respectively. For the comparison of differences between two variables, ANOVA Kruskal–Wallis with post hoc Dunn’s analysis was applied. The significance level of *p* < 0.05 was considered significant.

## Figures and Tables

**Figure 1 ijms-24-17242-f001:**
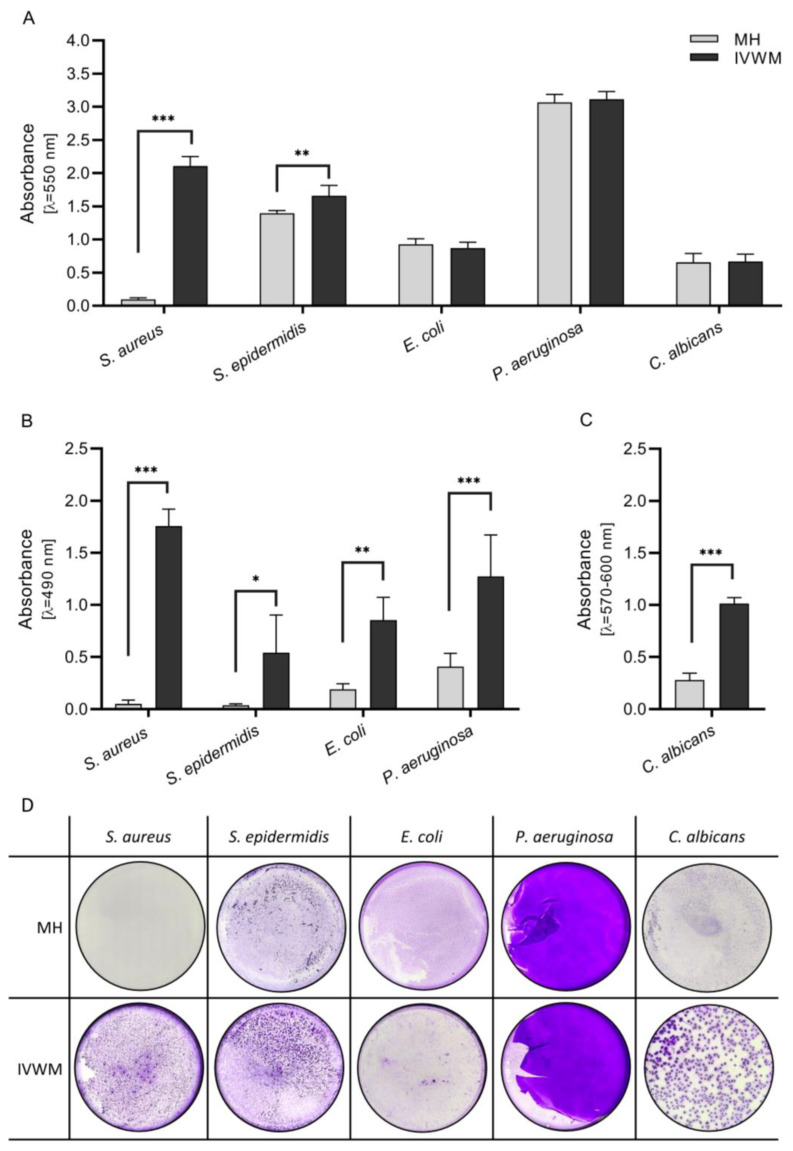
The average ability of the microorganisms to form biofilm in two media: Mueller–Hinton broth (MH) or In Vitro Wound Milieu (IVWM) measured with crystal violet staining (**A**,**D**), tetrazolium chloride staining (**B**), or resazurin staining (**C**). Standard deviations are marked with the error lines. Statistically significant differences are labeled as (***) *p* < 0.0001, (**) 0.0001 < *p* < 0.005, and (*) for 0.005 < *p* < 0.05.

**Figure 2 ijms-24-17242-f002:**
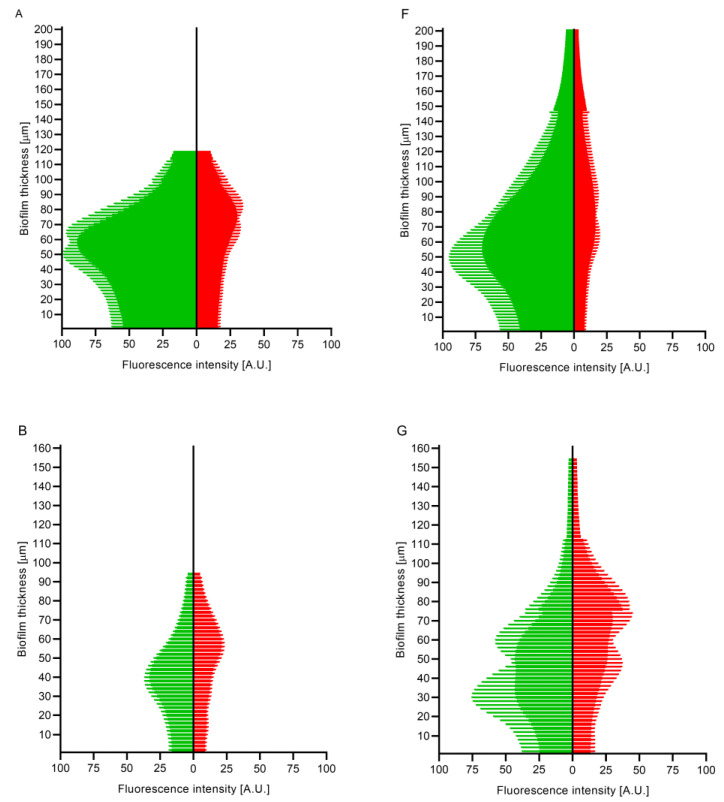
Cells’ distribution through the biofilm from its top to the bottom of *S. aureus* (**A**,**F**), *S. epidermidis* (**B**,**G**), *E. coli* (**C**,**H**), *P. aeruginosa* (**D**,**I**), and *C. albicans* (**E**,**J**) cultured in two media: Mueller–Hinton broth (MH) (**A**–**E**) or In Vitro Wound Milieu (IVWM) (**F–J**). Fluorescence intensity of propidium iodide (red color) and SYTO-9 (green color) represents damaged and non-damaged cells, respectively. The confocal microscope SP8, magnification 25×. A.U.—arbitrary unit.

**Figure 3 ijms-24-17242-f003:**
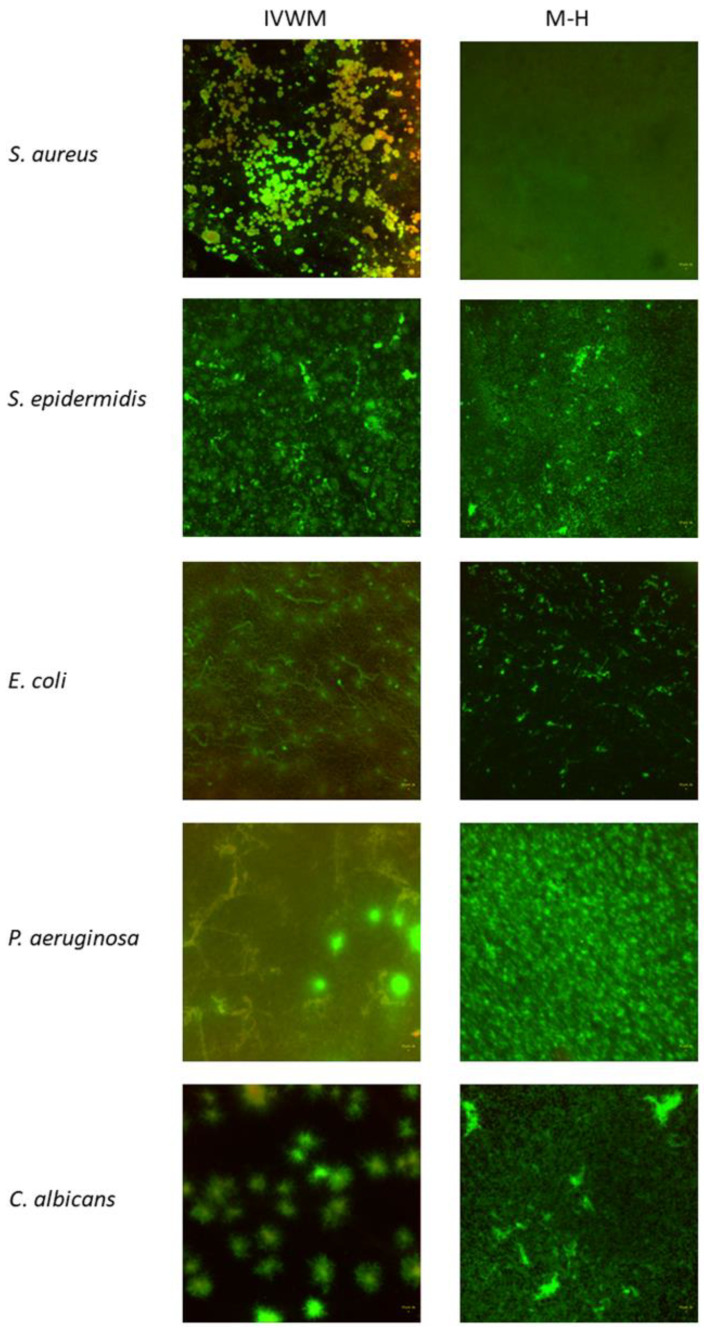
The differences in spatial distribution of biofilm-forming cells of tested pathogens cultured in IVWM or MH medium. Aerial view. The green shapes are cells of intact cell walls, while red/orange shapes are cells of damaged cell walls. Lumascope 620, magnification ×20. SYTO-9/Propionide Iodine fluorescent dyes.

**Figure 4 ijms-24-17242-f004:**
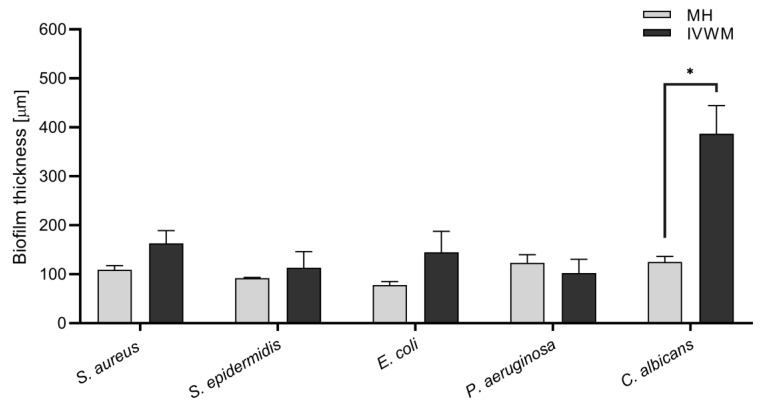
The average thickness (µm) of bacterial and fungal biofilm formed in two media: Mueller–Hinton broth (MH) or In Vitro Wound Milieu (IVWM), measured with confocal microscopy. Standard deviations are marked with the error lines. Statistically significant differences are labeled (*) for 0.005 < *p* < 0.05.

**Figure 5 ijms-24-17242-f005:**
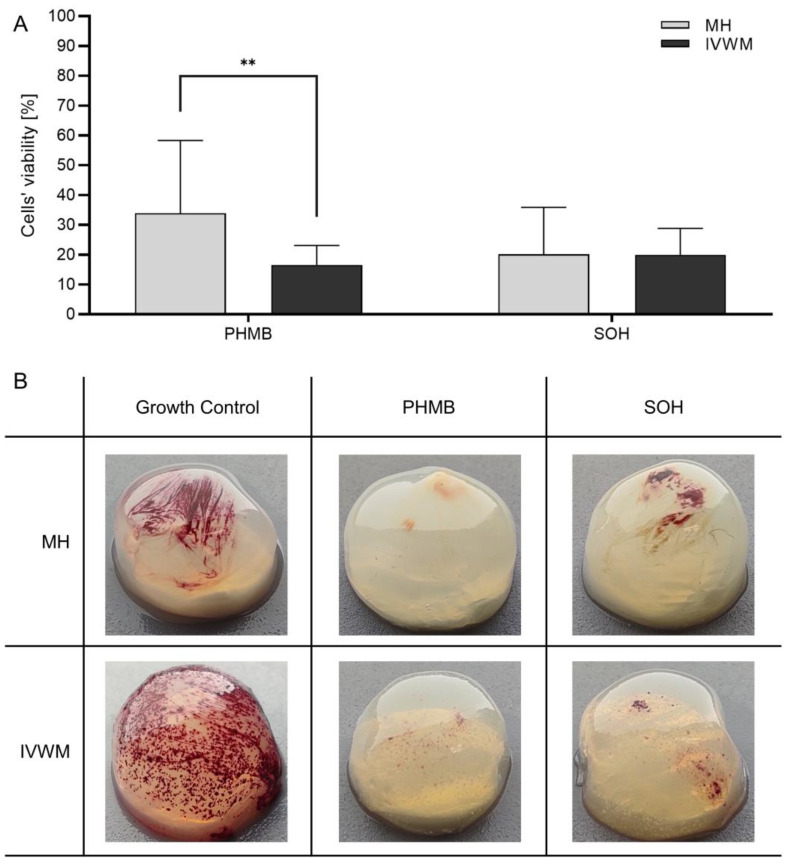
The average viability (%) of *Staphylococcus aureus* biofilm after treatment with polyhexamethylene biguanide hydrochloride (PHMB) or super-oxidized solution with hypochlorites (SOHs) in comparison to growth control, obtained in Mueller–Hinton broth (MH) or In Vitro Wound Milieu (IVWM) medium, measured in Cellulose-Based Biofilm (CBB) model (**A**) and bacterial cellulose carriers with stained biofilm (**B**). Standard deviations are marked with the error lines. Statistically significant differences are labeled as (**) for 0.0001 < *p* < 0.005.

**Table 1 ijms-24-17242-t001:** Minimal Inhibitory Concentration (MIC) of polyhexamethylene biguanide hydrochloride (PHMB), super-oxidized solution with hypochlorites (SOHs), and povidone-iodine (PVP-I), obtained in two media: Mueller–Hinton broth (MH) and In Vitro Wound Milieu (IVWM), presented in g/L; R—resistant within the tested concentration range.

	Minimal Inhibitory Concentration (mg/L)					
	PHMB		SOHs		PVP-I	
	MH	IVWM	MH	IVWM	MH	IVWM
*S. aureus*	2	2	R	R	1172	937.5
*S. epidermidis*	0.5	0.5	5	R	586	4688
*E. coli*	2	9	R	R	2344	9375
*P. aeruginosa*	8	63	R	R	9375	9375
*C. albicans*	0.5	0.5	R	R	2344	9375

**Table 2 ijms-24-17242-t002:** Minimal Biofilm Eradication Concentration (MBEC) of polyhexamethylene biguanide hydrochloride (PHMB), super-oxidized solution with hypochlorites (SOH) and povidone-iodine (PVP-I), obtained in two media Mueller–Hinton broth (MH) or In Vitro Wound Milieu (IVWM), presented in g/L; R—resistant within the tested concentration range; ND—non-detectable because of the insufficient growth.

	Minimal Biofilm Eradication Concentration (mg/L)					
	PHMB		SOHs		PVP-I	
	MH	IVWM	MH	IVWM	MH	IVWM
*S. aureus*	125	250	0.100	100	938	18,750
*S. epidermidis*	ND	ND	ND	ND	ND	ND
*E. coli*	63	125	100	100	9380	18,750
*P. aeruginosa*	500	250	R	R	18,750	18,750
*C. albicans*	15	500	100	R	9380	18,750

## Data Availability

The data presented in this study are openly available PPM repository at https://ppm.umw.edu.pl/info/researchdata/UMWda967363688c44ec99e2ab2d939da391/ (accessed on 15 November 2023).

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
