# Peer review of "Culture Shock: An Investigation into the Tolerance of Pathogenic Biofilms to Antiseptics in Environments Resembling the Chronic Wound Milieu"

_ijms, 2023, doi:10.3390/ijms242417242_

Round 1

Reviewer 1 Report

Comments and Suggestions for Authors

This study addresses an important issue for understanding and treatment of biofilms associated with chronic wounds. The authors compared biofilm formation and resistance to commonly used antiseptic compounds in five pathogens (Staphylococcus aureus, Staphylococcus epidermidis, E. coli, Pseudomonas aeruginosa and Candida albicans) in a standard lab medium (MH) and a medium resembling wound exudate (In Vitro Wound Milieu, IVWM). They report significant, in some cases multi-fold, differences in the level of biofilms and in bacterial tolerance to antiseptic compounds, thus uncovering the inadequacy of commonly used laboratory models. The relevance of this work is further enhanced by the experiments involving cellulose matrix, which resembles wound dressing materials, as a substrate for biofilm growth. Overall, the study is carefully planned and executed. One of its major limitations concerns testing of single representative strains of the five major wound-associated pathogens. Because of the strain-to-strain variability it is impossible to know whether or not study findings are applicable to other strains of these pathogens. However, this is an important start that timely calls to use biofilms models relevant to the disease.

I have no concerns about the experimental part but am concerned about the writing style.

Comments on the Quality of English Language

A major concern is that Discussion is unnecessarily verbose and reads like a review. Discussion needs to be significantly (approximately by half) shortened and focused on the study results. A reasonable start of the Discussion would be from line 295. Shortened general statements as well as the composition of the wound exudate can be moved to Introduction.

There are a number of trivial issues that require editing by a native English speaker, e.g., missing words, improper terms [e.g., 'altered cells' instead of 'damaged cells']), multiple spellings of the same abbreviations throughout the text, and poorly structured sentences in Discussion.

Author Response

Thank you for your positive feedback on our manuscript. We share your conviction regarding the importance of addressing critical issues in developing new and adequate models for testing the efficacy of antibiofilm agents. To assist you in reviewing our revisions, we have highlighted all changes made in response to your comments in green. Please note, however, that the sentences removed from the discussion section for brevity, as per your suggestion, are not highlighted.

Reviewer 2 Report

Comments and Suggestions for Authors

Authors investigated several strains of infective bacteria, with the main aim to see difference in formation of biofilm in context of used culturing media and antiseptic agent. Three agents were used, namely polyhexamethylene biguanide hydrochloride (PHMB), povidone-iodine (PVP-I) and super-oxidized solution with hypochlorites (SOH). And two different culturing media, namely In Vitro Wound Milieu (IVWM) and Mueller Hinton (MH) medium.

Authors report on substantial different behavior of strains depending on culture media such as biofilm development and susceptibility to antiseptic agents manifested as inhibitory concentrations. The manuscript is dealing with important and rarely reported scientific problems. I have only minor comments to this text.

PHMB is suspected cancerogenic – this information should be added to the introduction section of the manuscript

The common practice is to provide MIC values mg/L units to compare with available literature. (Table 1)

Polyamines display high dependance of activity in function of molar mass (e.g. 10.1021/acs.biomac.3c00139), what is the molecular mass of (PHMB) used in this study?

Author Response

Dear Reviewer 2, thank You for appreciating our manuscript. We are also convinced that it deals with important issues that should be addressed if we, as a scientific society, are going to create new, adequate models for testing efficacy of antibiofilm agents. For Your convenience all changes being result of answer to Your review are highlighted yellow.

Reviewer 3 Report

Comments and Suggestions for Authors

Dear authors, no concerns, I like the paper. 

Author Response

Dear Reviewer 3, thank You for appreciating our manuscript.
